# Whole-Transcriptome RNA Sequencing Reveals the Global Molecular Responses and CeRNA Regulatory Network of mRNAs, lncRNAs, miRNAs and circRNAs in Response to Salt Stress in Sugar Beet (*Beta vulgaris*)

**DOI:** 10.3390/ijms22010289

**Published:** 2020-12-30

**Authors:** Junliang Li, Jie Cui, Cuihong Dai, Tianjiao Liu, Dayou Cheng, Chengfei Luo

**Affiliations:** School of Chemistry and Chemical Engineering, Harbin Institute of Technology, Harbin 150086, China; xiaoga1206@163.com (J.L.); dch@hit.edu.cn (C.D.); sfangelos@163.com (T.L.); cfl7375@sina.com (C.L.)

**Keywords:** whole-transcriptome RNA-seq, microRNAs, lncRNA, circRNA, ceRNA, salt stress, sugar beet

## Abstract

Sugar beet is an important sugar-yielding crop with some tolerance to salt, but the mechanistic basis of this tolerance is not known. In the present study, we have used whole-transcriptome RNA-seq and degradome sequencing in response to salt stress to uncover differentially expressed (DE) mRNAs, microRNAs (miRNAs), long non-coding RNAs (lncRNAs) and circular RNAs (circRNAs) in both leaves and roots. A competitive endogenous RNA (ceRNA) network was constructed with the predicted DE pairs, which revealed regulatory roles under salt stress. A functional analysis suggests that ceRNAs are implicated in copper redistribution, plasma membrane permeability, glycometabolism and energy metabolism, NAC transcription factor and the phosphoinositol signaling system. Overall, we conducted for the first time a full transcriptomic analysis of sugar beet under salt stress that involves a potential ceRNA network, thus providing a basis to study the potential functions of lncRNAs/circRNAs.

## 1. Introduction

Crop growth and productivity is limited by high soil salinity, which affects plant physiological and biochemical processes [1,2]. Indeed, soil salinity leads to toxic levels of Na^+^ and Cl^−^ ions, increasing osmotic stress [3,4]. An excess of salt in the soil also reduces the water potential on the root surface and prevents water absorption [5]. Further, a high cytoplasmic Na^+^ concentration can disrupt the uptake of important ions such as potassium (K^+^), with adverse effects on enzyme catalytic activity and various metabolic pathways [6,7,8]. Osmotic and ionic stress can cause secondary stress in plants by accumulation of toxic compounds and reactive oxygen species (ROS) [4]. In response to these, plants have evolved mechanisms that include aquaporins [9], transporters [10] and various enzymes involved in free radical scavenging.

In addition to protein-coding RNAs, gene regulation by non-coding RNAs is also essential for plant adaptation to stressful conditions [11,12]. MicroRNAs (miRNAs) are small, endogenous, non-coding RNAs ranging from 19 to 24 nucleotides (nts) that regulate post-transcriptional events by cleaving their target mRNAs or by preventing translation [13]. Increasing evidence implicates miRNA in the gene regulation during seedling growth [14], floral development [15], male and female sterility [16,17] and responses to environmental stress [18]. Also, exported miRNAs have been shown to inhibit virulent gene expression by a plant fungal pathogen [19].

Long non-coding RNAs (lncRNAs) are another type of noncoding RNA that is longer than 200 nts [20]. In general, lncRNAs act via epigenetic modification, transcriptional regulation and post transcriptional regulation [21]. Recent genome-wide studies have revealed that they may be involved in plant stress; in *Populus trichocarpa*, a total of 504 lncRNAs were found to be drought responsive [22], and, in *Gossypium hirsutum*, 44 lncRNAs were responsive to salt stress [23]. Under salt stress, 185 differentially expressed lncRNAs have been reported in duckweed [24], whereas, in cotton, lncRNA973 regulates several genes, some of which are involved in oxygen scavenging [25]. In apples (*Malus domestica*), lncRNA MSTRG.85814.11 promoted the expression of *SAUR32*, which activated proton extrusion related to Fe-deficiency response [26]. 

Circular RNAs (circRNAs) are also non-coding RNAs with covalently closed circular structures, which makes them more stable than linear RNAs [27]. CircRNAs are rich in microRNA binding sites and can absorb cytoplasmic miRNA. Thus, they relieve the inhibition caused by miRNA and therefore increase gene expression [28,29]. High-throughput sequencing and bioinformatics analyses have shown that circRNAs have potent regulatory roles in plants [30]. In wheat, 62 circRNAs related to plant growth and survival were differentially expressed under dehydration [31]. In tomato (*solanum lycopersicum* L.), 163 circRNAs responded to low temperature stress [32]. In *Populus Euphratica* Oliv, 18 circRNAs participated in the stress response [33]. In *Arabidopsis*, overexpression of a circRNA derived from glycerol-3-p acyltransferase (ATS1), *Vv-circATS1*, improved cold tolerance [34]. Thus, circRNAs are involved in adaptive responses to abiotic stresses, but their role in the regulation of adaptive responses remains to be confirmed.

The competitive endogenous RNA (ceRNA) regulation hypothesis states that mRNA, lncRNA, circRNA and pseudogene transcripts modulate the stability or translational activity of target genes via competitive binding with miRNA to achieve post-transcriptional gene regulation [35]. The ceRNA mechanism invokes miRNA as the vehicle to mutually regulate RNAs, making coding and non-coding genes part of a large and sophisticated regulatory network within the whole transcriptome. From its initial applications in oncology, progress in third generation sequencing technology and bioinformatics analysis methods has expanded ceRNA research to other fields. For example, it is involved in the adaptation to low nitrogen in poplar [36], in the response to copper toxicity in *Citrus junos* [37] and in the response to heat stress in *Cucumis sativus* L. [38]. These results greatly enhance our understanding of the mechanisms of abiotic stress response in plants.

Sugar beet (*Beta vulgaris ssp. vulgaris*) is one of the most important sugar-yielding crops worldwide. As a recently domesticated crop, cultivated beets have inherited salt tolerance traits from their wild ancestor *Beta vulgaris ssp. maritima* (referred to as *B. maritima* or ‘sea beet’) [39]. However, research on sugar beet ncRNA is limited. Herein, we have used whole-transcriptome RNA sequencing of cultivar O68, a widely used sugar beet in China, to uncover the global molecular response to salt stress at both protein-coding RNAs (mRNAs) and non-coding levels (lncRNAs, miRNAs, and circRNAs). Degradome sequencing was also performed to validate the ceRNA networks. In summary, our objective was to provide a basis for studying the potential function of ceRNA in response to salt stress in sugar beet by (I) identifying how mRNA and ncRNA are involved in this response and (II) dissecting the underlying ceRNA regulatory networks. 

## 2. Results

### 2.1. Effects of Salinity on Sugar Beet Physiological Indices

We have shown previously that the relative germination rate of cultivar ‘O68′ under salt treatment was more than 90% and 70% in 200 and 300 mmol·L^−1^ NaCl, respectively [40], whereas seedling growth in the former case was even stronger than in the control group [40]. In the present study, seedlings exposed to 300 mmol·L^−1^ NaCl for 12 h showed morphological and physiological changes in both leaves and roots, such as clear wilting of leaves and color darkening of roots. We tested the tolerance to NaCl using physiological indices (Table 1), ‘ck’ represents the control group and ‘st’ represents the treatment group in this study. Salt stress reduced the relative water content and chlorophyll content in leaves. The soluble sugar content increased significantly, and the increase in leaves was greater than that of roots. Malondialdehyde content was slightly increased under salt stress in both leaves and roots. The content of proline in leaves increased significantly but not in roots. Moreover, the activity of peroxidase and superoxide dismutase was significantly enhanced under salt stress in both leaves and roots. The activity of catalase in leaves was significantly higher than that in roots and remained high under salt stress.

### 2.2. Global Response of mRNA to Salt Stress

A total of twelve transcriptome libraries were constructed from leaves and roots of six-leaf-stage seedlings which were either untreated (control) or treated with 300 mM NaCl. More than 1.5 billion raw reads were generated from RNA-seq with about 0.12 billion reads per sample. After quality control, 93.7% (1.4 billion) of the reads representing valid data were processed for further analysis. These reads were mapped on the beet genome with over 82.3% average mapping ratio. After quantification of gene expression levels as fragments per kilobase of transcript per million mapped reads (FPKM), 39,590 genes were identified as expressed genes. A Violin Plot shows the distribution of gene expression in each sample (Figure 1a). The image data indicated that gene expression levels were generally comparable among different time-point groups and these showed good repeatability. In leaves, 3578 differentially expressed mRNAs (DEmRNAs) (1180 up-regulated and 2398 down-regulated) were identified. A larger number, 4553 (1789 up-regulated, 2764 down-regulated), was identified in roots (Figure 1b and Appendix A), which suggests a more robust response to salt stress. Although some DEmRNAs expression was tissue specific, with 2436 and 3363 in leaves and roots, respectively (Figure 1c), 1084 were common to both (Figure 1c), which likely form the basic response to salt stress.

We used gene ontology (GO) annotation and Kyoto Encyclopedia of Genes and Genomes (KEGG) enrichment to explore DEmRNAs function (Figure 1d–g). GO annotation showed that the oxidation-reduction process was the dominant biological process (BP) in both leaves and roots (Figure 1d,e). However, in leaves, more DEmRNAs were annotated to response to cold and water deprivation, and flavonoid biosynthetic process under BP (Figure 1d). In contrast, in roots, those more abundant were defense response, signal transduction and response to abscisic acid (Figure 1e). For cellular component (CC), maximum DEmRNAs were annotated to chloroplast in leaves but plasma membrane in roots. In addition, a large amount of DEmRNAs belonging to the extracellular region and cell wall was found in both roots and leaves. In leaves, most DEmRNAs under molecular function (MF) were annotated to structural constituent of ribosome, RNA binding and endonuclease activity, while in roots were annotated to sequence-specific DNA binding, hydrolase activity and peroxidase activity. 

KEGG enrichment analysis showed enrichment in Ribosome (ko03010), Starch and sucrose metabolism (ko00500) and Isoflavonoid biosynthesis (ko00943) in leaves (Figure 1f), whereas Phenylpropanoid biosynthesis (ko00940), Plant hormone signal transduction (ko04075) and MAPK signaling pathway-plant (ko04016) were significantly enriched in roots (Figure 1g).

Identification of metabolic pathways affected by salt stress, and comparison of different organ responses, was performed with a MapMan analysis (Figure 1h-i and Appendix A). In general, salt stress led to up-regulation of DEmRNAs involved in glycolysis and nucleotide metabolism (degradation) in both leaves and roots (Figure 1h,i). In leaves, up-regulation was observed in DEmRNAs associated with starch and sucrose synthesis, amino acid catabolism, and some secondary metabolites (such as flavonoids and phenols), whereas those related to cell wall, lipid metabolism (FA synthesis), photosynthesis and amino acid synthesis were down-regulated (Figure 1h). In roots, in contrast, those related to cell wall and amino acid synthesis were up-regulated, whereas most related to secondary metabolism were down-regulated (Figure 1i). 

Regulation, cellular response, large enzyme families and transcription were used to evaluate metabolism-related DEmRNAs in detail (Appendix A). In both leaves and roots, metabolism for ABA (abscisic acid), ethylene and cytokinin (Appendix A), heat stress and Redox (Appendix A) were the most significantly up-regulated pathways. In leaves, many members of the cytochrome P450 family, peroxidases, UDP glycosyltransferases and the Glutathione-S-transferases family were up-regulated (Appendix A). In leaves, transcription analysis showed that members of the AP2-EREBP and HSF families were up-regulated in leaves (Appendix A). In roots, a large number of AP2-EREBP, bZIP, MYB and HB family members were significantly up-regulated (Appendix A). Thus, these metabolism pathways defined the differential response between leaves and roots to salt stress, even under the same genetic background. 

### 2.3. Global Response of lncRNA to Salt Stress

Apart from mRNAs, CNCI and CPC analysis identified 9076 potential lncRNAs from the RNA-seq data (Appendix A). Since no sugar beet lncRNA was included, all lncRNAs identified here are novel lncRNAs. When mRNA and lncRNA were compared, more than 60% of the mRNA was more than 1000 bp long, whereas more than 60% of the lncRNA was less than 500 bp long. Also, one to three exons were observed for a higher percentage of lncRNAs, which also had a shorter ORF length and lower FPKM value (Figure 2a–d). Finally, the number of DElncRNAs in leaves was 66 (32 up-regulated, 34 down-regulated), and this number was much higher in roots, 453 (218 up-regulated, 235 down-regulated) (Figure 2e–g), suggesting a more robust response to salt stress at the lncRNA level (Figure 2e). The general expression profiles of DElncRNAs (Figure 2f,g) show a similar proportion of up- and down-regulated.

To explore the potential functions of these DElncRNAs, their targeted mRNAs were predicted by cis-regulated analysis, 100 K upstream and downstream. In leaves, no suitable targeted mRNAs were found, but 55 were identified for 61 DElncRNAs in roots (Appendix A). The enriched GO terms of these targets are shown in Figure 2h, whereas enriched pathways indicated by KEGG enrichment analysis (Figure 2i) highlight targets such as Sulfur metabolism (ko00920), Pyrimidine metabolism (ko00240), Purine metabolism (ko00230) and Base excision repair (ko03410). 

### 2.4. Global Responses of circRNA to Salt Stress

In the leaf and root, 2625 potential circRNAs were identified (Appendix A), which showed no chromosome distribution preference (Figure 3a). Among these, 4.38, 10.52, and 85.10% belong to the intergenic region type, intron type and exon type, respectively (Figure 3b). In leaves, 13 DEcircRNAs were identified (eight up-regulated, five down-regulated), whereas 30 were found in roots (17 up-regulated, 13 down-regulated) (Figure 3d and Appendix A) and one was common to leaves and roots (Figure 3c). More DEcircRNAs were highly expressed in the treatment group, compared to leaves (Figure 4e) and roots (Figure 4f), suggesting that circRNAs are also involved in the response to salt stress.

### 2.5. Global Responses of microRNA to Salt Stress 

For miRNAs sequencing, we constructed twelve small RNA libraries from six-leaf-stage seedlings from four groups (treatment and control in roots and leaves) with three replicates. A total of 150 million raw reads with an average of 12.5 million per sample were generated. After filtering low-quality reads and removing sequences belonging to mRNA, repeat sequences and Rfam RNA tags, about 60 million reads with lengths 18 to 25 nt were selected as valid reads for further analysis. Finally, 494 mature miRNA candidates, including 165 known miRNAs (leaf: 153 and root: 130) and 329 novel miRNAs (leaf: 312 and root: 216) were identified in our small RNA sequencing data (Appendix A). These mature miRNAs can be mapped to 487 pre-miRNAs. In Dhom’s genome sequencing study for *B. vulgaris*, 522 ncRNA sequences were classified as pre-miRNAs [41], whereas 161 mature miRNAs from our small RNA sequencing can be mapped to these sequences (Appendix A). Those 21 nt long (37%) and 24 nt long (48%) account for the majority of candidate miRNAs, in agreement with the pattern of DICER-LIKE (DCL) cleavage in plant [42] (Figure 4a). Nucleotide composition of mature miRNAs indicated that both high-abundance miRNA clusters (21 nt or 24 nt) have a distinct sequence bias; 5′-uridine predominates in the first, and 5′-adenosine in the second (Figure 4b). This also has been observed in other plant species and is in agreement with the specificity of DCL restriction sites [42]. The distribution of identified miRNAs among different samples is shown in Figure 4c.

A total of 122 miRNAs showed significant changes in response to salt stress (Appendix A), 73 in leaves (20 up-regulated, 53 down-regulated) and 64 in roots (20 up-regulated, 44 down-regulated). However, although the DEmiRNAs expression seems tissue specific, some of these DEmiRNAs may belong to a same family. For instance, three MIR166 family members, mtr-miR166e-5p, mtr-miR166e-5p_2ss3AT10GT and mtr-miR166e-5p_L-2R+2, were differentially expressed in leaves, whereas mtr-miR166a and mtr-miR166e-5p did so in roots. The general expression profiles of these DEmiRNAs (Figure 4d,e) show obvious differences between ck and salt-treated samples and between root and leaf samples. About one-third were up-regulated in both leaves and roots under salt stress, whereas two-thirds were down-regulated. 

Unlike in animals, plant miRNAs have usually a perfect or near-perfect complementarity with their targets to induce target gene splicing and thus regulate gene expression; this feature can be used to predict plant miRNA targets. A total of 4150 target transcripts were identified for 432 miRNAs using the psRNATarget server and TargetFinder (Appendix A). About 87% of miRNAs obtained had more than one target (Appendix A). For example, a novel miRNA, PC-5P-3682_979, had 156 targets (Appendix A). This high number of target genes supports the conclusion that a single miRNA has the ability to cleave multiple targets. The GO annotation of DEmiRNAs shows the main targets in leaves (Figure 4f) and in roots (Figure 4g). The significantly enriched GO terms of these targets in leaves an in roots are shown in Appendix A, respectively. KEGG enrichment analysis showed that the significantly enriched pathways of these targets in leaves were Tyrosine metabolism (ko00350), Isoquinoline alkaloid biosynthesis (ko00950) and Sphingolipid metabolism (ko00600) (Appendix A), and Glyoxylate and dicarboxylate metabolism (ko00630), Carotenoid biosynthesis (ko00906) and Cysteine and methionine metabolism (ko00270) in roots (Appendix A).

### 2.6. CeRNA Regulatory Network in Response to Salt Stress 

To reveal the global regulatory network of mRNAs and ncRNAs under salt stress, a ceRNA network was constructed using DEmiRNAs, DEmRNAs, DElncRNAs, and DEcircRNAs based on ceRNA theory. After interaction analysis and correlation filtered, 15 DEmRNAs, 23 DElncRNAs and three DEcircRNAs were predicted to be targets of nine DEmiRNAs. These filtered genes were used to construct the salt stress response ceRNA network in leaves and roots (Figure 5). P threshold can be temporarily relaxed in some specific cases, such as the one-sided significance level was set at 0.2 in the study of Erlotinib alone or with Bevacizumab as first-line therapy in patients with lung cancer [43]. For leaves, since no ceRNAs had been obtained in leaves, the criteria of DEmRNA and DEncRNA were temporarily relaxed from *p* < 0.05 to *p* < 0.1 to explore potential ceRNAs, then qPCR was used to verify the expression correlation of all ceRNAs in subsequent experiments. Four novel miRNAs, PC-3p-125293_44, PC-5p-109182_51, PC-5p-7955_523 and PC-5p-3682_979, participated in the network, with the latter having an important role in both leaf and root under salt stress. The coding DEmRNAs of the ceRNA network in leaves included basic blue protein (*Bv1_023200_jmkt.t1*), uclacyanin-2 (*Bv6_144730_qgsa.t1*), stress protein DDR48 (*Bv1_011770_noyh.t1*), G-type lectin S-receptor-like serine/threonine protein kinase LECRK3 (*Bv4_071790_jfup.t1*) and tocopherol cyclase (*Bv2_032060_rrem.t1*). While the coding DEmRNAs of ceRNA network in roots included tubulin beta-1 chain (*Bv5_100510_ttjr.t1*), acid beta-fructofuranosidase (*Bv5_097930_juac.t1*), hypothetical protein LOC104907984 (*Bv5_124210_skcr.t1*), NAC domain-containing protein 21/22 (*Bv_000470_dyzk.t1*), NAC domain-containing protein 100 (*Bv5_114390_pjnp.t1*), uncharacterized protein CENP-C (*Bv9_220500_yszy.t2*), inositol hexakisphosphate and diphosphoinositol-pentakisphosphate kinase 2 (*Bv2_040500_uums.t1*), hypothetical protein BVRB_5g117790 (*Bv5_117790_mxei.t1*), malate synthase (*Bv5_110660_jxri.t1*), and hypothetical protein BVRB_9g213960 (*Bv9_213960_qddk.t1*).

### 2.7. Verification of the Cleavage of miRNA to ceRNAs by Degradome Sequencing 

Degradome sequencing was used to verify the cleavage of targets by miRNA. For leaves and roots, a respective degradome library was constructed and sequenced. A total of 51.4 million raw reads were generated. The clean tags were mapped to sugar beet reference genome and transcriptome data for identification of cleavage sites after consecutive steps of filtering. Based on the abundance of degradome tags at the target sites, cleaved targets were classified into five categories (0 to 4). The *p*-value here is not a screening parameter but for reference only; most p-values of Category 0 and Category 1 were less than 0.05 (Appendix A). A total of 752 miRNA–mRNA pairs were identified for 141 miRNAs and 444 transcripts (including 406 mRNAs and 38 lncRNAs) and six pairs of miRNA–mRNA, which were used to build the ceRNA network. Some representative cleavage sites are shown as target plots (T-plots) (Figure 6). Interestingly, degradome sequencing confirmed that a single miRNA can act on multiple targets, e.g., mtr-miR408-3p_L-1R+1 can cleave both *basic blue protein* (*bv1_023200_jmkt*) and *uclacyanin-2* (*bv6_144730_qgsa*) (Figure 6b,c). In addition, different miRNAs from the same family can work together to regulate the same genes, e.g., *basic blue protein* (*bv1_023200_jmkt*) was cleaved by gma-miR408a-3p_L-1R+5, mtr-miR408-3p_L-1R+1 and mtr-MIR408-p3_2ss18GT19GA (Figure 6a,c,d). Combining the expression patterns of DEmRNAs and DEmiRNAs, 16 and 17 negative correlations in expression were identified in miRNA–mRNA pairs in leaves and roots, respectively, where 11 pairs were confirmed by degradome sequencing (Table 2).

### 2.8. qRT-PCR Validated Expression Correlation between miRNAs and ceRNAs under Salt Stress

To confirm the results of RNA-seq and to validate the expression correlation of miRNAs and their targets, all components of the ceRNA network were selected for qRT-PCR detection (Figure 7). The qRT-PCR results agree well with RNA-seq data regarding expression trends. However, qPCR fold-change of some genes was different from the sequencing data, especially for ncRNA, possibly because the different principles underlying these two detection methods. The expression of all miRNAs was negatively correlated with ceRNAs, whereas those of ceRNAs were positively correlated. For example, gma-miR6300_L-1R+1 was up-regulated and all of its targets were down-regulated; PC-5p-109182_51 was down-regulated and all of its predicted targets were up-regulated. This result not only demonstrates that the RNA-seq data in this study is reliable, but also validates the negatively correlated expression between miRNAs and ceRNAs. No expression was detected for two down-regulated lncRNAs in the ST group (MSTRG.30182.1 and MSTRG.28802.1), possibly because of low abundance.

## 3. Discussion

Salt stress threatens the normal growth and development of plants by directly inducing osmotic imbalance, ion injury and reactive oxygen species (ROS). Thus, response to salt stress involves a variety of physiological and biochemical processes, such as osmotic regulation, ion balance or reactive oxygen scavenging. These are regulated by the expression of a large number of genes. Although miRNAs, lncRNAs and circRNAs have been identified in plants, there are no studies of ncRNAs in sugar beet. In this study, DEmRNAs, DElncRNAs, DEcircRNAs and DEmiRNAs were identified in leaves and roots under salt stress. These results support the ceRNA hypothesis which has progressively wider acceptance since it was first proposed [35]. These RNAs compete for binding to common MREs and thus are mutually regulated, with implications for posttranscriptional gene regulation in physiological and pathophysiological processes [44]. From this perspective, we used our sequencing data in sugar beet to construct a ceRNA regulatory network to elucidate the mechanisms underlying tolerance to salt. The fact that the number of salt stress-response genes (including mRNA and ncRNA) in roots was higher than that in leaves, and the functional divergence observed in these two organs, suggests that roots have a more robust and complex response to salt. 

### 3.1. Analysis of Salt Stress Response in Sugar Beet Leaves

Functional analysis of differentially expressed genes (DEGs) in leaves showed that leaf cells appear to regulate gene expression with the maintenance of photosynthesis as the core. Some genes directly involved in photosynthesis were significantly up-regulated under salt stress. For example, NAD(P)H-quinone oxidoreductase chain 2 (*Bv7_171100_eawo.t1*) involved in NADH dehydrogenase (ubiquinone) activity, *proton gradient regulation protein 5* (*Bv3_048340_odem.t1*) involved in photosynthetic electron transport in photosystem I or *STN7 protein kinase* (*Bv6_155590_ugjw.t1*) involved in the regulation of light reaction. In addition, some genes located in the chloroplast that play a protective role in photosynthesis were also significantly up-regulated under salt stress. For example, *choline monooxygenase* (*Bv6_146100_wdro.t1*) and *betaine aldehyde dehydrogenase* (*Bv5_116230_ntjn.t1*), which are involved in the synthesis of betaine [45]; glutathione peroxidase (*Bv2_044150_kdwp.t1*), *ferrochelatase 2* (*Bv2_033330_nrat.t1*) and *soul heme-binding family protein* (*Bv4_094190_wifn.t1*) involved in heme biosynthesis in non-photosynthetic tissues and induced by oxidative stress in photosynthetic tissues to supply heme for defensive hemoproteins. Studies of *rhodiola crenulata* have shown that glutathione peroxidase can physically interact with specific proteins in multiple pathways which participate in photosynthesis, respiration and stress tolerance [46]. Further, *alpha amylase* (*Bv1_021820_ccxs.t1*) was significantly up-regulated under salt stress, which suggests faster starch degradation by mesophyll cells. This may increase soluble sugar content and provide energy to regulate osmotic pressure and respond to salt stress. The increase in soluble sugar content in leaves also supports this hypothesis (Table 1).

Copper ion plays an irreplaceable role in chlorophyll synthesis, stabilization and photosynthesis; thus, regulating metal homeostasis in chloroplasts is extremely important [47]. Plants may have to redistribute a limited copper ion population in leaves, since salt stress results in reduced transpiration and hindered transport of copper ions from the roots. Indeed, we found that *basic blue protein* (*Bv1_023200_jmkt.t1*) and *uclacyanin-2* (*Bv6_144730_qgsa.t1*) were down-regulated by the mtr-MIR408-p3_2ss18GT19GA-mediated ceRNA network (Figure 8), which may be related to copper distribution. This is in agreement with our previous studies on the proteome of sugar beet under salt stress revealing up-regulation of plastocyanin in leaves [48], consistent with copper ions being preferentially supplied to photosynthesis. Other studies have shown that the expression of miR408 was up-regulated under abiotic stress in Arabidopsis thaliana [49] and that the overexpression of miR408 enhanced drought tolerance in chickpea (*Cicer arietinum* L.) [50].

Under salt stress, increased membrane permeability and disruption of Na^+^/K^+^ homeostasis is enabled by α-tocopherol (α-T) by increased unsaturation of the membrane [51], where tocopherol cyclase may play an important role. Our results show that PC-5p-3682_979-mediated ceRNAs up-regulated the expression of tocopherol cyclase (*Bv2_032060_rrem.t1*) (Figure 8). Under salt stress, the content of tocopherol was significantly decreased in beet leaves, while the content of MDA did not change significantly (Table 1). We hypothesized that sugar beets protect their membrane lipids from peroxidation by consuming tocopherol. The same Figure shows that the ceRNA network revealed up-regulation of ABA induced stress protein DDR48 (*Bv1_011770_noyh.t1*) and receptor-like protein kinase 1 (RPK1) (*Bv4_071790_jfup.t1*) located in the endomembrane system; these may be involved in the response to salt stress through ABA signaling and phosphorylation pathways. In *Arabidopsis thaliana*, RPK1 is involved in regulating the shutdown of guard cells that is key for plant stress response [52].

Moreover, a number of genes associated with ABA and ethylene metabolism and signal transduction were up-regulated in leaves: *protein phosphatase 2C* (*Bv8_197440_jnqo.t1*), *zeaxanthin epoxidase* (*Bv_003210_fkxn.t1*), *carotenoid cleavage dioxygenase* (*Bv5_107230_qtix.t1*) and ERF3 (*Bv2_031130_cghd.t1*), *1-aminocyclopropane-1-carboxylate oxidase* (*Bv8_185300_wndh.t1*). Indeed, Nicotinamide adenine dinucleotide (NAD) responds to salt stress by affecting the accumulation of ABA and proline [53] and up-regulation of NAD(P)H-quinone oxidoreductase and ABA-related genes and the increase in proline content support this conclusion. 1-aminocyclopropane-1-carboxylate oxidase (ACO) is a key enzyme in the ethylene biosynthesis pathway. The more than 20-fold change of carotenoid cleavage dioxygenase has received the most attention, suggesting that it may play an important role in the response to salt stress. These results suggest that leaves are regulated to maintain photosynthesis under salt stress (Figure 8).

### 3.2. Analysis of Salt Stress Response in Sugar Beet Roots

In contrast to what we observed in leaves, roots showed a broader mobilization under salt stress. Functional analysis of DEGs in roots indicated that root cells shift resources from growth and development to survival by adjusting several processes, e.g., glycometabolism and metabolism of fatty acids and energy and secondary metabolism. Sucrose is the end product of photosynthesis, yielding hexoses (Hexes) necessary to generate energy and synthesize diverse biomolecules. A significant up-regulation of sucrose synthases (*Bv7_163460_jmqz.t1* and *Bv8_190960_nnjy.t2*) was detected in root cells under salt stress. Sucrose metabolism has been associated with high sensitivity of development to abiotic stress in plants, since a reduction in hexose triggers a downstream response to stress [54]. Also, pressure can affect the expression of acid beta-fructofuranosidase, involved in the metabolic signaling of primary metabolism and defense response [55,56]. Our results show that gma-miR6300_L-1R+1-mediated ceRNAs down-regulate the expression of *Bv5_100510_ttjr.t1* and *Bv5_097930_juac.t1*. *Bv5_100510_ttjr.t1* (Figure 8). These genes encode tubulin beta-1 chain protein mostly expressed in endodermal and phloem cells of primary roots. The role of microtubules as signal transducers for plant stress response has been reported [57]. *Bv5_097930_juac.t1* encodes acid beta-fructofuranosidase which participates in the degradation of sucrose and is associated to hexose signals. Interestingly, a series of glycolytic-related genes were up-regulated under salt stress in roots, such as *hexokinase* (*Bv9_224680_aote.t1*), *phosphofructokinase* (*bv9_212010_mhum.t1*) and *pyruvate kinase* (*Bv7_160680_tjji.t1* and *Bv2_034650_iitw.t1*), which are the rate-limiting enzymes in glycolysis (Figure 8). Pyruvate dehydrogenase (*Bv_004020_oxcs.t1*) and NADP-dependent malate dehydrogenase (*Bv8_197290_sfck.t1* and *Bv_006710_gkqg.t1*) were also up-regulated in roots. Further, PC-5p-109182_51-mediated ceRNAs up-regulate the expression of *Bv5_110660_jxri.t1* and *Bv9_213960_qddk.t1*. *Bv5_110660_jxri.t1* encodes malate synthase, involved in the conversion of fatty acids to carbohydrates and important in maintaining gluconeogenesis [58]. Expression of *phosphoenolpyruvate carboxykinase* (*Bv6_152970_ygjo.t1*) was reduced by 20-fold under salt stress. Thus, salt stress may weaken the gluconeogenic pathway, and acetyl-coa and oxaloacetic acid is probably provided to the tricarboxylic acid cycle by other pathways. These results suggest that salt stress increases energy metabolism in root cells.

NAC domain-containing proteins are one of the largest plant-specific transcription factor families. Proteins in this family are characterized by a highly-conserved N-terminus, known as the DNA-binding NAC domain (NACs), and a highly divergent C-terminal region containing the transcription regulatory region. Besides their critical functions in plant growth and development [59], NACs have also been implicated in the response to abiotic and biotic stresses [60]. We show here that two NAC family members, NAC domain-containing protein 21/22 (*Bv_000470_dyzk.t1*) and NAC domain-containing protein 100 (*Bv5_114390_pjnp.t1*), are down-regulated by mtr-miR164d-mediated ceRNAs under salt stress (Figure 8). We confirmed cleavage of these two genes by mtr-miR164d by degradation sequencing. This down-regulation may activate multiple downstream genes, which is in agreement with a *Nicotiana benthamiana* study infected by beet necrotic yellow vein virus [61]. PC-5P-3682_979 regulates the expression of tocopherol cyclase in leaves, where it may perform a different function. In roots, a PC-5P-3682_979-mediated ceRNA seems to be involved in the phosphoinositol signaling system since it down-regulated inositol hexakisphosphate and diphosphoinositol-pentakisphosphate kinase 2 (*Bv2_040500_uums.t1*) and F-box/kelch-repeat protein At3g06240-like (*Bv5_117790_mxei.t1*) (Figure 8).

In addition, the ceRNA network also revealed the down-regulation of two functional unknown proteins uncharacterized protein CENP-C (*Bv9_220500_yszy.t2*) and hypothetical protein LOC104907984 (*Bv5_124210_skcr.t1*), which may play roles in response to salt stress by unknown pathway in sugar beet (Figure 8). Additionally, more remarkable, many AP2-EREBP, bZIP and MYB family transcription factors were significantly up-regulated under salt stress, such as ERF73 (*Bv7_163350_ryod.t1*), DREBP1 (*Bv3_066590_ignp.t1*), TGA3 (*Bv1_021380_gmre.t1*), ABSCISIC ACID-INSENSITIVE 5 (*Bv7_159570_afnu.t1*), MYB39 (*Bv8_181400_zguf.t1*), etc. It has been reported that the DREB transcription factor is involved in the salt stress response of tomatoes and *arabidopsis thaliana* [62] and that the overexpression of DREB transcription factors can increase the salt tolerance of potatoes (*solanum tuberosum*) [63]. These transcription factors may play a crucial regulation role in the response to salt stress in roots. Similar to leaves, a large number of genes related to ABA and ethylene metabolism were also significantly upregulated in roots under salt stress (Figure 8), for example, *PP2C 8* (*Bv3_052290_rcph.t1*) and *PP2C 24* (*Bv7_173230_chqm.t1*), *1-aminocyclopropane-1-carboxylate synthase* (*Bv1_004040_gcto.t1*), and so on. Further, peroxidase 5-like (*Bv4_079730_uxri.t1*), glutathione S-transferase (*Bv4_095800_rgfa.t1*), and some cell wall-related genes such as pectinesterase 11 (*Bv1_006810_mswa.t1*) and pectin acetylesterase 8 (*Bv5_098360_owsf.t1*) were also significantly up-regulated under salt stress (Figure 8). The above results support the idea that an extensive metabolic regulation was carried out in roots in favor of cell survival under salt stress (Figure 8).

Overall, these results indicate that miRNAs–lncRNAs-circRNAs–mRNAs networks are involved in regulating gene expression to modify growth, improve photosynthesis, adjust plasma membrane permeability, promote glycometabolism and energy metabolism, regulate transcription factor, and participate in the phosphoinositol signaling system during adaptation to salt stress. 

## 4. Materials and Methods 

### 4.1. Plant Cultivation and Treatments

Sugar beet ‘O68′ was used in this study. Seeds from our laboratory (Heilongjiang, China) were soaked in water for ten hours, sterilized in 0.1% (*v*/*v*) HgCl_2_ for 10 min, washed repeatedly with distilled water and germinated on wet filter paper in a germination box at 26 °C for 2 days. After germination, seedlings were transferred to plastic pots (43.5 cm × 20 cm × 14 cm, 10 plants per pot) filled with quarter-strength Hoagland solution. The germinating seeds were cultivated under 16/8 light photoperiod at 24 °C (day)/18 °C (night) in a phytotron (Friocell 707, Germany). Six-leaf-stage seedlings (28 days) were treated with 300 mmol NaCl for 12 h and untreated plants served as controls. 

### 4.2. Measurement of Physiologic Indicators and Harvest

Fresh samples were used to measure physiologic indicators. All of the physiological indicators listed in Table 1 were carried out in accordance with the Experimental Guidelines for Plant Physiology [64]. Samples were collected from leaves and roots (about 6 cm from the root tip) as shown in Appendix A for sequencing. Variability was minimized by constructing the library with samples taken from six different plants. All collected samples were immediately frozen in liquid nitrogen for half an hour and stored at −80 °C until further use.

### 4.3. RNA Extraction, Library Preparation, and RNA Sequencing

A total of 0.5 g of leaves, or roots, was taken from six plants and mixed to obtain each sample. Total RNA was extracted from the roots and leaves of ck and st samples by TRIZOL (Invitrogen, CA, USA) according to the manufacturer’s instructions. Quantity and purity of total RNA were analyzed using Bioanalyzer 2100 and RNA 6000 Nano LabChip Kit (Agilent, CA, USA) with RIN number > 7.0. High-quality total RNA from each sample was used to construct libraries for RNA-seq and small RNA sequencing. Total RNA from leaves and roots was mixed for degradome sequencing. Library construction and sequencing were conducted by LC Sciences (Hangzhou, China).

For mRNA, LncRNA and circRNA sequencing, about 5 µg of total RNA from each sample was used to deplete ribosomal RNA according to the Ribo-Zero™ rRNA Removal Kit (Illumina, San Diego, USA). After removing ribosomal RNAs, RNAs were fragmented by divalent cations under a high temperature. Cleaved fragments were reverse-transcribed to create the cDNA, which was used to synthesize U-labeled second-stranded DNAs with *E. coli* DNA polymerase I, RNase H and dUTP. There were three biological replicates per group and a total of 12 libraries (leaf_ck, leaf_st, root_ck, root_st). The average insert size in the final cDNA library was 300 bp (±50 bp). Finally, we performed paired-end sequencing on an Illumina Novaseq™ 6000 (LC Bio, China) following the vendor’s recommended protocol. For small RNA sequencing, 12 small RNA libraries (leaf_ck, leaf_st, root_ck, root_st) were constructed according to the TruSeq Small RNA Sample Preparation (Illumina, San Diego, CA). The small RNA fragments were finally sequenced by Illumina HiSeq2500 platforms. The raw data of RNA-seq (BioProject ID: PRJNA666384) and small RNA sequencing (BioProject ID: PRJNA666142) were deposited in the NCBI SRA database.

### 4.4. Read Mapping and Transcriptome Assembly

For RNA-Seq data, the raw reads obtained from sequencing were processed to remove primer/adaptor contamination, low-quality bases and undetermined bases using Cutadapt [65]. FastQC (http://www.bioinformatics.babraham.ac.uk/projects/fastqc/) was used to verify sequence quality. Bowtie2 [66] and Hisat2 [67] were used to map reads to the reference genome RefBeet-1.2 (http://bvseq.boku.ac.at/Genome/Download/RefBeet-1.2/). The mapped reads of each sample were assembled using StringTie [68]. 

### 4.5. Differentially Expressed mRNA and Bioinformatics Analysis

All transcriptomes from all samples were merged to reconstruct a comprehensive transcriptome using Perl scripts, after which StringTie [68] and edgeR [69] estimated the expression levels of all transcripts. StringTie was used to perform the expression level of mRNAs by calculating FPKM (fragments per kilobase of transcript per million mapped reads). The differential expression analysis was performed using edgeR R-packages. The *p*-value < 0.05 and |log_2_ (fold change)| > 1 were defined as the cutoff criteria for DEGs. UpSetR was used to visualize differences in gene expression patterns between groups [70]. By default, 0.000001 was added to all FPKM values to avoid computing log (0).

All transcripts were annotated by conducting Blastx similarity searches against NCBI non-redundant protein databases (Nr) with an E-value threshold of 10^−5^. All transcripts were blasted to the Gene Ontology database and calculated gene numbers for each term. A hypergeometric test was used to find significantly enriched GO terms in the input gene list [71]. The Kyoto Encyclopedia of Genes and Genomes (KEGG) [72] was used to perform pathway enrichment analysis. Mapman software was used to analyze DEGs (http://mapman.gabip.org, version: 3.6.0). 

### 4.6. Identification and Analysis of lncRNAs

Transcripts that overlapped with known mRNAs, and transcripts shorter than 200 bp, were discarded during the identification of novel lncRNAs. We used CPC [6] and CNCI [7] to predict transcripts with coding potential. All transcripts with CPC score < 0.5 and CNCI score < 0 were removed and those remaining were considered to be lncRNAs. StringTie was used to perform expression level for mRNAs and lncRNAs by calculating FPKM. The differentially expressed lncRNAs (DE lncRNAs) were selected with |log_2_ (fold change)| > 1 and *p*-value < 0.05 by edgeR. To explore the function of lncRNAs, we predicted their cis-target genes. LncRNAs may play a cis role acting on neighboring target genes. In this study, coding genes in 100,000 upstream and downstream were selected by Python script. Then, we showed functional analysis of the target genes for lncRNAs by using the BLAST2GO [73]. Significance was expressed as a *p*-value < 0.05.

### 4.7. Identification and Analysis of CircRNAs

In this study, Bowtie2 [66] and Hisat2 [67] were used to map reads to the genome of sugar beet. The remaining reads (unmapped reads) were still mapped to genome using tophat-fusion [74]. CIRCExplorer2 [75,76] and CIRI [77] were used to de novo assemble the mapped reads to circular RNAs at first; then, back splicing reads were identified in unmapped reads by tophat-fusion [74]. All samples were generated unique circular RNAs. Differentially expressed circRNAs (DE circRNAs) were extracted with |log_2_ (fold change)| >1 and with statistical significance (*p*-value < 0.05) by R package–edgeR.

### 4.8. Identification and Analysis of miRNAs

The raw reads obtained from sequencing were subjected to ACGT101-miR (LC Sciences, Houston, Texas, USA) to remove adapter dimers, junk, low complexity, common RNA families (rRNA, tRNA, snRNA, snoRNA) and repeats. As no sugar beet miRNA was recorded in miRBase 21.0, the filtered unique reads with lengths between 18 and 25 nt were mapped to viridiplantae mature miRNAs and precursors by BLAST search to identify conserved and previously reported miRNAs, and the mapped pre-miRNAs were further aligned against the *B. vulgaris* genomes to determine their genomic locations using Bowtie [78]. Only miRNAs matched to known miRNAs with no more than three mismatches in the miRBase database, and whose precursors could fold into stem-loop structures, were considered to be known miRNAs. Subsequently, the unmapped sequences were blasted against the *B. vulgaris* genomes to identify potential novel miRNAs, and the hairpin structures-containing sequences were predicted from the flanking 200 nt sequences using the mfold software [79]. A global normalization method was used to correct copy numbers among different samples [80]. The latter method can eliminate the effect of sequencing discrepancy on the calculation of small RNA expression. Therefore, the calculated gene expression can be directly used to compare the difference of gene expression among samples. Only miRNAs with Fold Change 1.5 and *p*-value < 0.05 were classified as DE miRNAs.

### 4.9. Degradome Sequencing Validation of the Cleavage of miRNA to Target Genes

For degradome library construction, about 20 μg of total RNA from both treatment and control groups were pooled together to generate a mixed-library for leaf and root. The library construction was performed as described with some modifications [81]. Single-end sequencing (36 bp) of the purified cDNA libraries was performed on an Illumina Hiseq2500 at the LC-BIO (Hangzhou, China) following the vendor’s recommended protocol. Raw sequencing reads were obtained using Illumina’s Pipeline v1.5 software following sequencing image analysis by Pipeline Firecrest Module and base-calling by Pipeline Bustard Module. The raw reads were processed to remove low-quality reads, reads with ‘N’ and any reads with adaptor and primer contamination using ACGT101-DEG (Lc-bio Sciences, Houston, USA). The clean reads were used to identify the degraded fragments of mRNA after removing the reads that can be annotated into a different ncRNA database. Potential miRNA editing sites were identified using the small RNA sequencing data by CleaveLand 3.0 [82]. The identified sites were classified into five categories (0, 1, 2, 3 and 4) based on read abundance at that position as reported previously [83]. Based on the signatures and abundance in the sugar beet RNA-seq data, T-plots were built for high efficiency analysis of potential miRNA targets.

### 4.10. Construction and Analysis of ceRNAs Regulatory Network

In plants, there are two patterns of complementarity between miRNA and their target gene mRNA; one leads to complete target degradation, whereas, in the other, miRNA is sequestered and therefore inactivated by miRNA target mimics [84]. Based on the ceRNA hypothesis, a ceRNA network can be built by predicting miRNA-binding RNA. In this study, miRNA–mRNA and miRNA–ncRNA pairs were predicted by TargetFinder (https://github.com/carringtonlab/TargetFinder) and Ssearch36 (36.3.6) (https://fasta.bioch.virginia.edu/fasta_www2/fasta_down.shtml), respectively. In brief, (I) the bulge must be on the ncRNA and located in the middle of the miRNA, (II) except for the position in the middle of miRNA, a maximum of 4 mismatches are allowed and there cannot be more than two continuous mismatches and (III) no bulge is allowed in a non-middle position. A ceRNA network was visualized with Cytoscape software (version 3.6.1) [85]. 

### 4.11. qRT-PCR Validation of DEmRNAs, DElncRNAs, and DEmiRNAs

qRT-PCR was performed to validate the expression level of DEmRNAs, DE lncRNAs, DE circRNAs and DE miRNAs. Total RNA and small RNA was extracted from leaves and roots samples using an GeneJET Plant RNA Purification Mini Kit (Thermo Scientific, Cat#K0801, Lithuania) and mirVana™ miRNA Isolation Kit (Invitrogen, Cat#K157001, USA), respectively. Firststrand cDNA was synthesized from 1 μg total RNA with a High-Capacity cDNA Reverse Transcription Kit (Applied Biosystems, Cat#4368814, USA) for qPCR of mRNA and miRNA, and with the Verso cDNA Synthesis Kit (Thermo Scientific, Cat#AB1453A, Lithuania) for qPCR of lncRNA and circRNA. The qPCR reactions were performed on CFX Real-time PCR system (BIO-RAD, USA) using iTaq Universal SYBR^®^ Green Supermix (BIO-RAD, Cat#1725125, USA) following the manufacturer’s instructions. For miRNAs, the primers including miRNA-specific stem-loop RT, forward primers and universal reverse primer for the selected miRNAs were designed according to Kramer [86]. The 5S RNA + U6 gene was used as endogenous controls. For mRNA, lncRNA and circRNA, PP2A + UBQ5 and PP2A + 25S RNA genes were used as endogenous controls for root and leaf, which were selected from 15 candidate genes evaluated using Bestkeeper [87], NormFinder [88] and GeNorm [89]. To avoid non-specific amplification, a melting curve was carried out for each PCR product. The expression level of the miRNAs in different samples was calculated by the comparative 2^−△△CT^ method. For the data of the physiological parameters and qPCR analysis, the mean and SD were calculated from three repeats per treatment, and differences were analyzed by Duncan’s multiple range test (*p* < 0.05) and an independent samples *t*-test (*p* < 0.05). All primers are shown in Appendix A.

## 5. Conclusions

We have used whole-transcriptome RNA-seq to reveal the divergent response to salt stress in leaves and roots, identifying 3578 and 4553 DEmRNAs, 66 and 453 DElncRNAs, 73 and 64 DEmiRNAs and 13 and 30 DEcircRNAs, respectively. Our results suggest that specific lncRNAs and circRNAs function as ceRNAs in response to salt stress. GO term, KEGG pathway and Mapman analyses showed that competitive lncRNA/circRNA/mRNA-miRNA regulatory networks are implicated in copper redistribution, plasma membrane permeability, glycometabolism and energy metabolism, NAC transcription factor and the phosphoinositol signaling system. In a nutshell, the metabolic regulation of leaves revolves around ensuring photosynthesis to obtain the energy necessary to cope with environmental pressure whereas roots obtain energy by enhancing glycometabolism and fatty acid metabolism to cope with these environmental changes. Many transcription factors such as MYB, DREBP, NAC and plant hormones such as ABA and ethylene are also involved in the salt stress response. Overall, these results lay the foundation for studying lncRNAs/circRNAs response to salt stress in different organs of sugar beet.

## Figures and Tables

**Figure 1 ijms-22-00289-f001:**
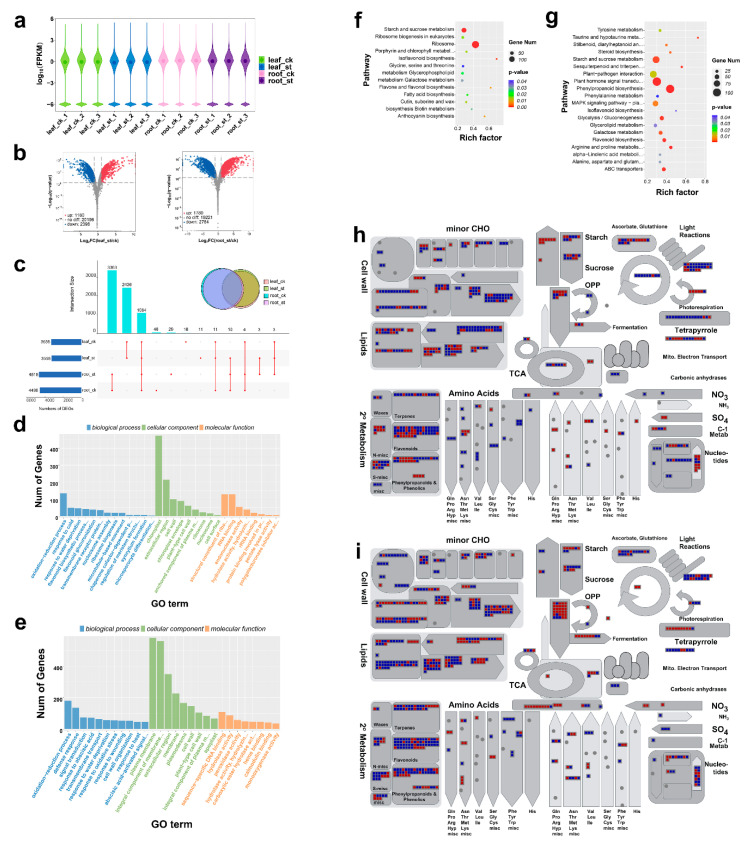
Identification and analysis of differentially expressed mRNAs (DEmRNAs) under salt stress. (**a**) Violin Plot of gene expression patterns for each sample, with origin representing the median; ck represents the control group and ST represents the treatment group. (**b**) Volcano Plot of log2 FC(ST/CK) of leaves and roots mRNAs. (**c**) Visualization of DEmRNAs in each sample, where the intersection size represents those common to all samples, intersection size stands for the common DEmRNAs among different samples. (**d**–**e**) gene ontology (GO) annotation in leaves (**d**) and roots (**e**). (**f**,**g**) Kyoto Encyclopedia of Genes and Genomes (KEGG) enrichment in leaves (**f**) and roots (**g**). (**h**,**i**) Mapman analysis of metabolism in leaves (**h**) and in roots (**i**), where each square represents mapping to the metabolic pathway and color represents up-regulation (red) or down-regulation (blue).

**Figure 2 ijms-22-00289-f002:**
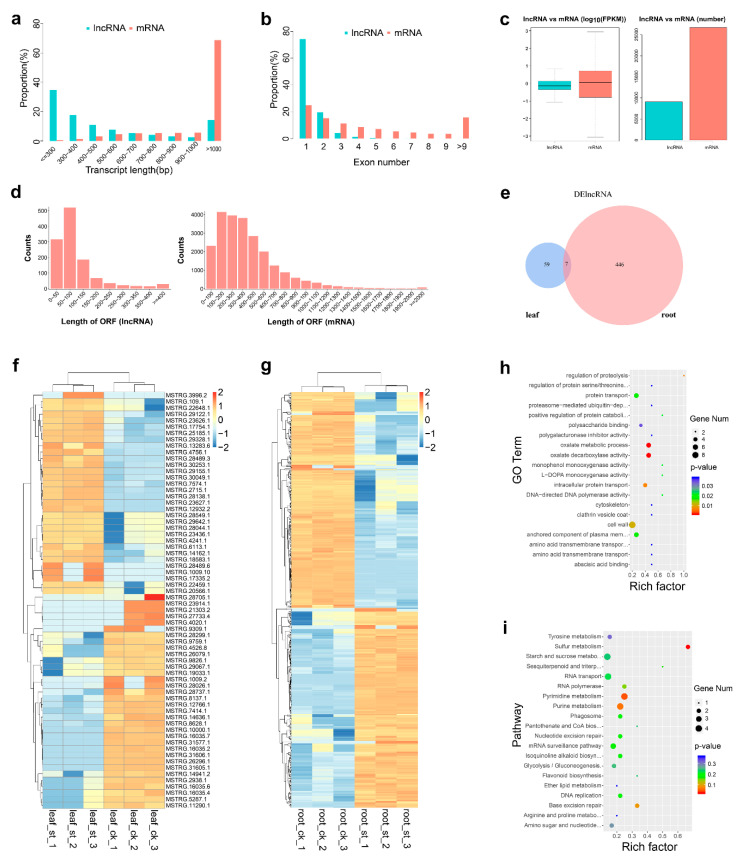
Identification and analysis of DElncRNAs under salt stress. (**a**–**d**) Comparison of transcript length, exon number, fragments per kilobase of transcript per million mapped reads (FPKM) value, numbers and ORF length in lncRNA versus mRNA. (**e**) Venn diagram of DElncRNAs in leaves and roots. (**f**,**g**) Heat map of DElncRNAs in leaves (**g**) and in roots. (**h**) GO enrichment of targets of DElncRNAs in roots; (**i**) KEGG enrichment of targets of roots DElncRNAs.

**Figure 3 ijms-22-00289-f003:**
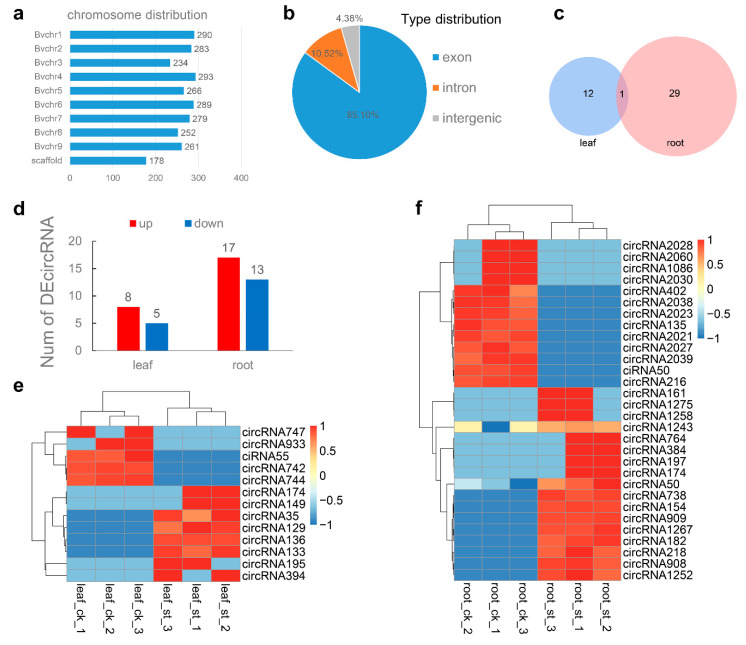
Identification and analysis of DEcircRNAs under salt stress. (**a**,**b**) All identified circRNAs chromosome distribution (**a**) and type distribution (**b**). (**c**) Venn diagram of DEcircRNAs in leaves and roots. (**d**) Number of DEcircRNAs in leaf and root. (**e**,**f**) Heat map of DEcircRNAs in leaves (**e**) and in roots (**f**).

**Figure 4 ijms-22-00289-f004:**
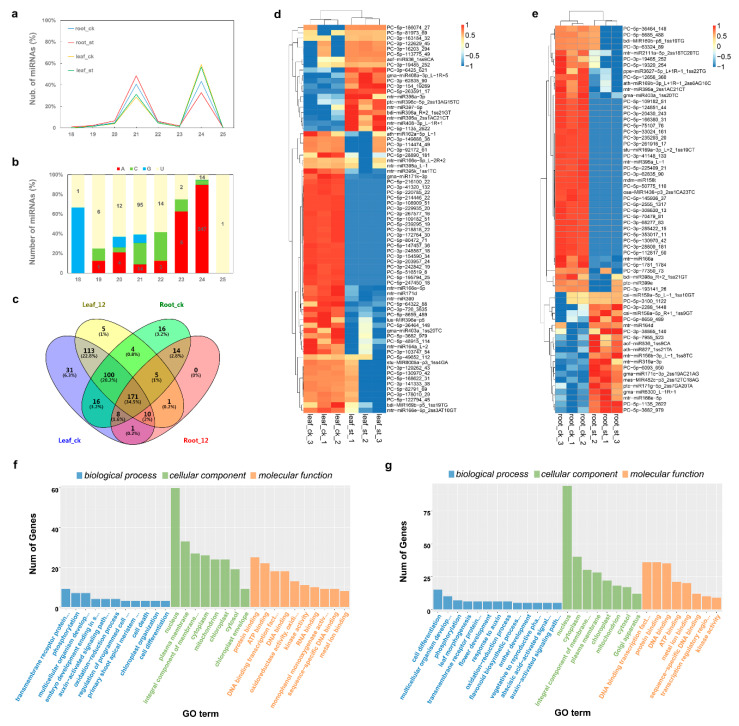
Identification and analysis of DEmiRNAs under salt stress. (**a**,**b**) Identified miRNAs length distribution (**a**) and 5′ nucleotide composition (**b**). (**c**) Venn diagram of identified miRNAs in each sample. (**d**,**e**) Heat map of DEmiRNAs in leaves (**d**) and in roots (**e**). (**f**,**g**) GO annotation of targets of DEmiRNAs in leaves (**f**) and in roots (**g**).

**Figure 5 ijms-22-00289-f005:**
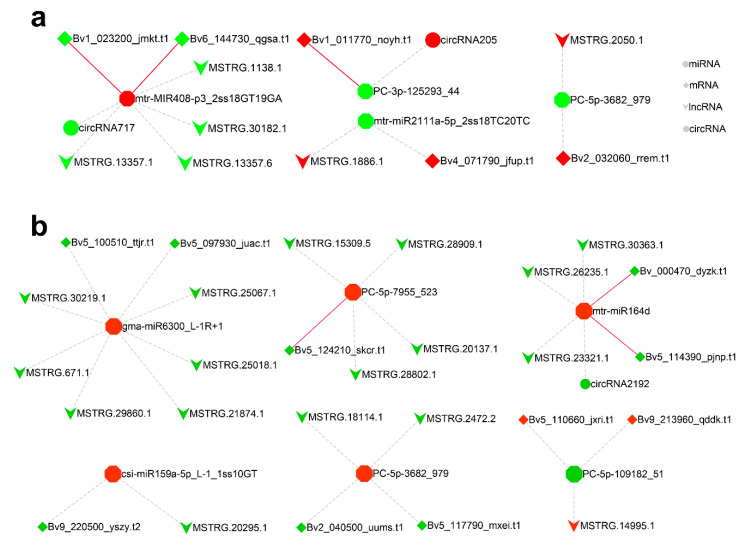
Salt stress response ceRNA network constructed with DEmRNAs, DElncRNAs and DEmiRNAs in leaves (**a**) and roots (**b**). The color represents up-regulation (red) or down-regulation (green). Red solid lines represent regulation verified by degradome sequencing.

**Figure 6 ijms-22-00289-f006:**
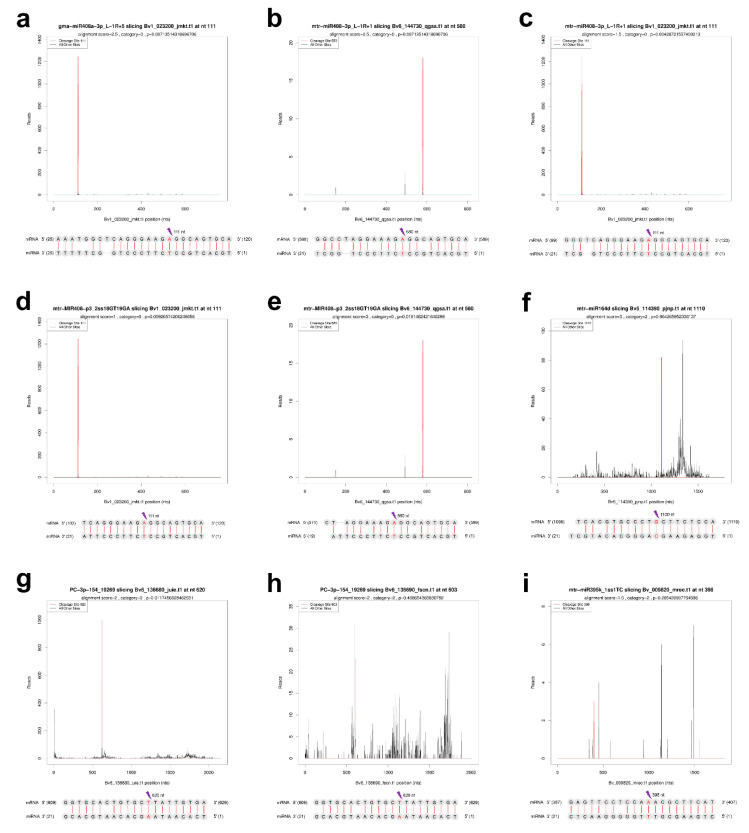
T-plots and alignments of miRNA–mRNA pairs validated by degradome sequencing. (**a**) *bv1_023200_jmkt.t1* cleaved by gma-miR408a-3p_L-1R+5; (**b**) *bv6_144730_qgsa.t1* cleaved by mtr-miR408-3p_L-1R+1; (**c**) *bv1_023200_jmkt.t1* cleaved by mtr-miR408-3p_L-1R+1; (**d**) *bv1_023200_jmkt.t1* cleaved by mtr-MIR408-p3_2ss18GT19GA; (**e**) *bv6_144730_qgsa.t1* cleaved by mtr-MIR408-p3_2ss18GT19GA; (**f**) *bv5_114390_pjnp.t1* cleaved by mtr-miR164d; (**g**) *bv6_136680_juie.t1* cleaved by PC-3p-154_19269; (**h**) *bv6_136690_fscn.t1* cleaved by PC-3p-154_19269; (**i**) *bv_009820_mrec.t1* cleaved by mtr-miR395k_1ss1TC. The ‘lightning’ shape indicates the position of the cleavage site.

**Figure 7 ijms-22-00289-f007:**
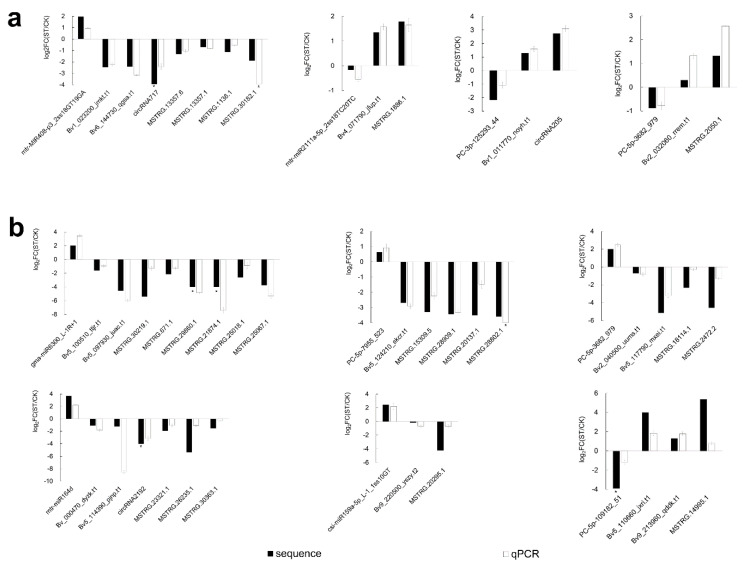
qRT-PCR analysis of all 50 components of the ceRNA network under salt stress in (**a**) leaves and (**b**) roots. FC(ST/CK) represents fold changes of relative expression levels between ST and CK, log2FC(ST/CK) is from the mean of three replicates, bars stand for ±SD, asterisk (*) indicates that the gene is not detected in ST and its value is artificially set to −4 instead of calculating log2(0).

**Figure 8 ijms-22-00289-f008:**
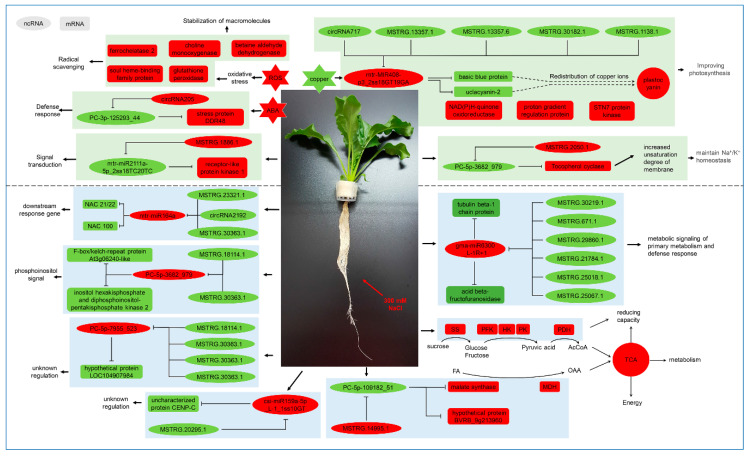
Proposed model of ceRNA network that regulates response to salt stress in sugar beet. SS: sucrose synthase, PFK: fructose phosphate kinase, HK: hexokinase, PK: pyruvate kinase, PDH: pyruvate dehydrogenase, MDH: malic dehydrogenase. Red filling represents up-regulation, whereas green filling represents down-regulation.

**Table 1 ijms-22-00289-t001:** Physiological parameters under salt stress.

Physiological Indices	ck (Leaf)	st (Leaf)	ck (Root)	st (Root)
Relative water content (%)	90.17 ± 0.06 *	87.73 ± 0.15 *	-	-
Chlorophyll (mg·g^−1^ FW)	2.104 ± 0.069 *	1.759 ± 0.096 *	-	-
Tocopherol (μg·g^−1^ FW)	223.22 ± 14.70 *	164.50 ± 10.87 *	-	-
Soluble sugar (mg·g^−1^ FW)	0.3664 ± 0.0186 *	0.5835 ± 0.0121 *	0.2792 ± 0.0067 *	0.3309 ± 0.0151 *
MDA (nmol·g^−1^ FW)	8.3427 ± 0.1464	9.4161 ± 0.8450	6.6839 ± 0.0845	8.1476 ± 0.5915
Proline (μg·g^−1^ FW)	1.4131 ± 0.1915 *	4.0466 ± 0.0335 *	0.5334 ± 0.0213	0.5897 ± 0.0563
POD activity (U·g^−1^ FW)	48.14 ± 1.26	44.43 ± 1.29	121.85 ± 13.85 *	206.85 ± 8.66 *
SOD activity (U·g^−1^ FW)	24.57 ± 0.54 *	41.07 ± 6.28 *	24.62 ± 1.33 *	64.09 ± 2.49 *
CAT activity (U·g^−1^ FW)	568.78 ± 2.71	569.36 ± 2.89	36.45 ± 1.28 *	21.07 ± 1.15 *

Data are means ± SD (*n* = 3). * indicate *p* < 0.05 by Tukey test. FW: fresh weight; MDA: malonaldehyde; POD: peroxidase; SOD: superoxide dismutase; CAT: catalase.

**Table 2 ijms-22-00289-t002:** Negative regulation DEmiRNA–mRNA pairs in leaves and roots under salt stress.

miR_name	up/down	Transcript	up/down	Tissue	Degradome Detection
mtr-miR408-3p_L-1R+1	up	Bv6_144730_qgsa.t1	down	leaf	Y
mtr-miR408-3p_L-1R+1	up	Bv1_023200_jmkt.t1	down	leaf	Y
gma-miR408a-3p_L-1R+5	up	Bv1_023200_jmkt.t1	down	leaf	Y
PC-3p-154_19269	up	Bv6_136680_juie.t1	down	leaf	Y
PC-3p-154_19269	up	Bv6_136690_fscn.t1	down	leaf	Y
mtr-miR395a_L-1	down	Bv_009820_mrec.t1	up	leaf	Y
mtr-miR395k_1ss1TC	down	Bv_009820_mrec.t1	up	leaf	Y
gma-miR403a_1ss20TC	down	Bv4_094840_csri.t1	up	leaf	Y
mtr-miR395a_L-1	down	Bv_010680_tsww.t1	up	leaf	N
lus-MIR396e-p5	down	Bv1_015660_nghm.t1	up	leaf	N
PC-3p-726_3835	down	MSTRG.2715.1	up	leaf	N
PC-5p-109182_51	down	Bv4_078360_ftyu.t1	up	leaf	N
PC-5p-36464_148	down	Bv1_011830_oexd.t1	up	leaf	N
PC-5p-3682_979	down	Bv5_114160_hsdp.t1	up	leaf	N
PC-5p-3682_979	down	Bv8_193090_kary.t1	up	leaf	N
PC-5p-49652_112	down	MSTRG.29155.1	up	leaf	N
csi-miR156a-5p_R+1_1ss9GT	up	Bv6_136190_cygi.t1	down	root	Y
mtr-miR164d	up	Bv5_114390_pjnp.t1	down	root	Y
PC-5p-7955_523	up	Bv5_124210_skcr.t1	down	root	Y
gma-miR6300_L-1R+1	up	Bv5_097930_juac.t1	down	root	N
gma-miR6300_L-1R+1	up	Bv5_100510_ttjr.t1	down	root	N
gma-miR6300_L-1R+1	up	Bv9_208900_gijo.t1	down	root	N
PC-5p-3682_979	up	Bv1_012390_utfq.t1	down	root	N
PC-5p-3682_979	up	Bv1_017030_kxfa.t1	down	root	N
PC-5p-3682_979	up	Bv5_117790_mxei.t1	down	root	N
PC-5p-3682_979	up	Bv6_132150_dsqx.t1	down	root	N
PC-5p-3682_979	up	Bv6_135930_aphq.t1	down	root	N
PC-5p-3682_979	up	Bv8_193090_kary.t1	down	root	N
ptc-miR399e	down	Bv5_121080_kswa.t1	up	root	N
PC-5p-109182_51	down	Bv5_110660_jxri.t1	up	root	N
PC-5p-109182_51	down	Bv9_213960_qddk.t1	up	root	N
PC-5p-130970_42	down	MSTRG.29564.3	up	root	N
PC-5p-36464_148	down	MSTRG.25036.1	up	root	N

## Data Availability

All sequencing data was deposited in the NCBI Short Read Archive (SRA) database under the BioProject ID: PRJNA666384 (RNA-seq) and BioProject ID: PRJNA666142 (miRNA and degradome sequencing). Relevant supporting data can be found within the article and additional files.

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
