# Peer review of "Whole-Transcriptome RNA Sequencing Reveals the Global Molecular Responses and CeRNA Regulatory Network of mRNAs, lncRNAs, miRNAs and circRNAs in Response to Salt Stress in Sugar Beet (Beta vulgaris)"

_ijms, 2020, doi:10.3390/ijms22010289_

Round 1

Reviewer 1 Report

This study by Li et al., with an aim to investigate the competitive endogenous RNA regulation hypothesis is presented very nicely and I really enjoyed reading it. This study not only describes the scientific discussion very well, but also has identified some new molecular components in the field of salt stress tolerance and presents a holistic view of interactions among various kind of molecules.

Few minor comment are given below,

Legends in figures text are hard to understand, such as figure 1c, f, g, and 2a. Please elaborate all the abbreviations in the figure legends as well, e.g. what is leaf_ck.

Figure 4 is missing

Author Response

Legends in figures text are hard to understand, such as figure 1c, f, g, and 2a. Please elaborate all the abbreviations in the figure legends as well, e.g. what is leaf_ck.

Response: We have adjusted the font size of some legends, and the DPI of all figures to 600. We have also added full names for abbreviations where they first appear in the article

Figure 4 is missing

Response: Figure 4 has been added to its correct location.

Reviewer 2 Report

The manuscript of Li et al., describes a comprehensive analysis of the whole transcriptome of sugar beet aiming to identify transcriptomic responses to salt stress. Apart from mRNA sequencing, this study also includes sequencing and identification of small and regulatory RNAs and suggest a holistic network of transcripts. This study sheds light in responses under salt stress of an important crop.

The manuscript is descriptive and easy to follow and the analyses well conducted.

I just have a few minor comments that would improve the manuscript.

Line 61; please include the Latin name of apples

Line 73; participated -->participate

Line 75; please delete "can"

Line 94; please italicize the species

Line 193; please delete "were"

Line 196; were-->was

Lines 447-451; Please include one or two references to support the sentences.

Lines 533-534 and 676; The given Bioproject IDs do not give any result in NCBI SRA. Is there possibly a typo or are these deposited data under embargo until publication?

Author Response

Line 61; please include the Latin name of apples

Response: "Malus domestica" has been added.

Line 73; participated -->participate

Response: Corrected.

Line 75; please delete "can"

Response: We have deleted this word.

Line 94; please italicize the species

Response: We have corrected it to italics, as required.

Line 193; please delete "were"

Response: We have deleted this word as required.

Line 196; were-->was

Response: We have corrected this word as required.

Lines 447-451; Please include one or two references to support the sentences.

Response: We have added two references here, as required.

Lines 533-534 and 676; The given Bioproject IDs do not give any result in NCBI SRA. Is there possibly a typo or are these deposited data under embargo until publication?

Response: The data has not been released yet. It will be released in due course.

Reviewer 3 Report

Salinization has been a global threat to crop production. The research by Junliang Li et al. is a descriptive study largely missing validations of the observed changes. Still, it could be a valuable contribution to our understanding of the salinity stress response in crop plants.

However, the present state of the manuscript is far from satisfactory. There are multiple typos (including figures, chromosome discribution), and the manuscript requires extensive language editing and proofreading. Figure 4 is missing, the authors did not remove text from the template (L490-495), and there are also mistakes in units (e.g. mMol.L-1). 

Major issues:

1) Results - Authors should present a full description of the salinity-stress experiment, including some measurable evidence of the stress response on plant physiology. That is presently missing.

2) Discussion of the experimental setup and employed techniques should be limited to results/discussion and excluded from Materials and Methods.

3) Figures are illegible (Figure 1c, f, g, h; Figure 2a,b; Figure 6). Authors should increase the resolution and eliminate small fonts. 

4) Statistics - authors should not manipulate with statistical significance threshold to "provide more results" (L284).

5) This research would significantly benefit from supporting analyses. For instance, soluble sugars or tocopherol are easily determined. The authors should at least provide a more detailed comparison with the previous studies. The salinity stress response has been studied in model plants, yet the comparison to that research is limited. Authors could highlight the similarities and differences found here.

Minor issues:

1) Table 1 does not contain results of a correlation analysis, the legend should be modified. 

2) "Specific part of roots" used for sampling are not visualized in Fig S3. Further, it is not clear why the leaf tissue was sampled as illustrated. The leaf discs were pooled into 0.5 g aliquotes, and thus the benefits of spatial distribution are lost. Could you please explain why this additional mechanical stress was beneficial for the analysis?

Author Response

Response: The manuscript has been edited very thoroughly by EditSprings services https://www.editsprings.com/). The text has been significantly trimmed and we hope the text is now more readable. Figure 4 has been added to its correct location. The text in the template has been deleted. Units have also been fixed.

Major issues:

  • Results - Authors should present a full description of the salinity-stress experiment, including some measurable evidence of the stress response on plant physiology. That is presently missing.

Response: Thank you for your advice. “2.1 Effect of salinity on sugar beet physiological indices” has been added to describe this content.

  • Discussion of the experimental setup and employed techniques should be limited to results/discussion and excluded from Materials and Methods.

Response: Thank you for your advice. We have moved this content from Materials and Methods to Results (2.1 Effect of salinity on sugar beet physiological indices).

  • Figures are illegible (Figure 1c, f, g, h; Figure 2a,b; Figure 6). Authors should increase the resolution and eliminate small fonts.

Response: Thank you for your advice. We have corrected the font size in Fig. 1 and Fig. 2. For Figure 6, since all t-plot files obtained by analysis of degradation sequencing are in JPG format, it is difficult to modify the font size, but we have ensured that these images can be clearly displayed after enlargement. Resolution of all figures has been adjusted to 600 DPI as required. All figures can be viewed clearly after zooming. 

  • Statistics - authors should not manipulate with statistical significance threshold to "provide more results" (L284).

Response: We agree that statistical significance is important in the analysis of high-throughput sequencing results. However, the P-value only indicates the probability of something happening; it does not add additional properties to the data. We usually define P < 0.05 as the significance threshold to eliminate the risk of "false positives" as much as possible, but this value is relative and can be changed.

For example, some clinical studies use a P lower than 0.01,0.001 or even 0.0001 to reduce this risk. In a similar way, we argue that the P threshold can be temporarily relaxed in some specific cases, such as in the study of ceRNA. Due to the lack of related studies many mechanisms need to be explored, and we are aware that relaxing the P value may lead to more possible "false positives".

Similarly, fold change (FC)>2 is a sort of standard, but results are too limited using this value when studying protein or miRNA difference expression levels, therefore this threshold is usually defined as 1.5. In circRNA studies, difference expression circRNA may be as high as 10 even though the threshold is defined as 2. These thresholds are tools to help filter the data. Although we may define FC > 2 as a statistically significant, this may not correspond to biological significance. For example, here we assumed that only genes with FC > 2 were likely to participate in the salt stress response, but those with FC 1.5 may also be involved.

Based on these considerations, we relaxed the screening criteria to obtain more possibilities when constructing the ceRNA network. Of course we have ensured that our results are reliable by carrying out qPCR verification of all ceRNAs and miRNAs obtained to reduce false positives as much as possible. We are aware that the qPCR results show FC value differences in a few genes that do not reach the difference standard defined traditionally, but these are showed in our results and not hidden. Future research can use our results to study other ceRNAs of interest. 

  • This research would significantly benefit from supporting analyses. For instance, soluble sugars or tocopherol are easily determined. The authors should at least provide a more detailed comparison with the previous studies. The salinity stress response has been studied in model plants, yet the comparison to that research is limited. Authors could highlight the similarities and differences found here.

Response: Thank you for your advice. We have added some physiological indices in Results 2.1. These physiological indices were tested prior to sequencing. Tocopherol content was not measured at the time of sampling because it is not a conventional index for stress assessment. Therefore, we plan to use the cryopreserved (-80) sample for measurement. However, due to the time needed to purchase vitamin E standard, we need an extra week to complete this experiment. In addition, we added a comparison for the response of some genes in model crops.

Minor issues:

  • Table 1 does not contain results of a correlation analysis; the legend should be modified.

Response: We have corrected the legend as “Negative regulation DE miRNA-mRNA pairs in leaves and roots under salt stress.”  Notice that the original Table 1 is now Table 2.

  • "Specific part of roots" used for sampling are not visualized in Fig S3. Further, it is not clear why the leaf tissue was sampled as illustrated. The leaf discs were pooled into 0.5 g aliquotes, and thus the benefits of spatial distribution are lost. Could you please explain why this additional mechanical stress was beneficial for the analysis?

Response: The issue of sampling for sequencing is an interesting one. Our lab has been using high-throughput sequencing technology since 2014. At first, due to lack of experience and lack of rigorous sampling, large amounts of differentially expressed genes (DEG) were obtained in early transcriptome studies (A total of 22,000 have been detected by high-throughput sequencing while more than 12,000 genes were belonging to DEGs). This represents either many false positives or incorrect sampling, rather than being related to the treatment applied.

In our experience, sampling should be as "standardized" as possible to eliminate unexpected sampling-related differences. There are large differences between individual plants. For example, seeds grown under the same conditions present different germination times or seedlings grow faster than others. To eliminate these differences, we selected plants with the same growth potential and extracted RNA after mixed sampling of multiple plants.

However, a problem appeared when sampling: a piece of leaf weighs about 0.5 ~ 0.7 g (the third pair of euphylla), therefore the combining six plants weighed far more than the samples uesd to extract the RNA (0.5 g), which means that sampling a whole leaf cannot be performed. Our sampling method (Figure S3) covers all differentiated leaf cells, e.g., epidermal, mesophyll, guard and vascular cells.

Regarding spatial distribution are Lost, let's look at this example first: Acid beta-fructofuranosidase and sucrose synthase (SS) are two sucrose metabolizing enzymes. The former is responsible for unloading exogenous sucrose (from leaf) at the cell wall, and the latter is responsible for intracellular sucrose metabolism. Since the root cells cannot fix carbon, we assumed that the gene encoding for acid beta-fructofuranosidase should be highly expressed gene in cells. However, the FPKM of this gene was only 2 in the control group, and less than 1 after salt stress, while SS was as high as 300 in the control group and 1,700 after stress. Is that difference in expression of these two genes real? We think not, because acid beta-fructofuranosidase is usually expressed in phloem cells, while SS is expressed in all cells, However, our sampling method for root heavily diluted the expression of acid beta-fructofuranosidase. Of course, this is just a possibility that has to be confirmed, but nevertheless is shows that sampling can affect results greatly. After stress, epidermal cells, guard cells, vascular bundle cells, root cap cells, cambium cells and other differentiated cells may have their own specific transcriptional differences. These differences are better explored by single-cell sequencing technology, but herein transcriptional changes in leaf or root were only studied at the macro level. We are aware that this miss spatial distribution information, but in the M&M section we clearly indicate the sampling location which should make our experiment highly reproducible, where results due to different culture batches should cancel out.

For root sampling, the root part shown in Fig. S3, 6 cm from the root tip. Here sampling problem is the same as in the leaves. Root weight of one plant was over 1 g, and a standardized sampling method was required to extract RNA from 0.5 g samples. Since the root tip is the most active for water and nutrient absorption, cells at the root tip were selected. As shown in Figure S3, the area 6 cm upward from the root tip of each plant was taken and mixed.

The sugar beet is a biennial crop whose first year is devoted to vegetative growth, building lush foliage and large root tubers. Sugar beet roots are up to 3 m long under normal growth conditions (Beet Physiology; we used the Chinese version edited by Wenzhang Qu (ISBN75388-1391-8)). When we sampled at 3 pair of euphylla stage, the average length of the root system was about 40 cm. Due to sufficient nutrition in hydroponics the lateral roots developed rapidly. There was no significant difference in length and diameter between the taproot and some lateral roots at 3 pair of euphylla stage. In summary, our sampling method covers almost all the root tips of a sample.

Needless to say, we will be glad to hear of other ideas about sampling, which we can implement in future studies.

Round 2

Reviewer 3 Report

The authors have addressed at least some of my concerns.  

  • Figures - resolution is still too low for the converted PDF, but that could be solved with the original files in 600 dpi
  • Statistics -
    "For leaves, the criteria of DEmRNA and DEncRNA were temporarily relaxed from p <0.05 to p <0.1 to obtain more results." 
    - I do understand the meaning of the p-value and that the thresholds can be manipulated. However, the whole dataset should be processed with the same settings, or the threshold manipulation should be justified and explained in more detail, including references that would support that. You could easily lower the threshold to 0.5 and get "more results". 
  • Addition experiments 
     "However, due to the time needed to purchase vitamin E standard, we need an extra week to complete this experiment."

    The authors require one more week to provide supporting experiments that would provide real evidence for the transcriptomic-based predictions. I believe that this extension should be allowed. 

  • Spatial distribution - the authors have probably misunderstood my comment - by pooling samples from different locations, the spatial distribution within the leaf tissue is lost. The same results would have been obtained by grinding the whole leaf tissue in liquid nitrogen and aliquoting. The additional stress (which occurs within µs, see e.g. https://science.sciencemag.org/content/361/6407/1068?iss=6407) that could be relevant for the presented results would be eliminated. However, this is only a minor issue and does not need to be addressed.

    Please, extend the figure legend of Fig. S3 to make it more self-explanatory - the fact that the whole root segment was collected is not intuitive 

Author Response

  • Figures - resolution is still too low for the converted PDF, but that could be solved with the original files in 600 dpi

Response: Thank you for your suggestion. We have submitted a 600DPI file for each figure separately in the submission system.

  • Statistics -
    "For leaves, the criteria of DEmRNA and DEncRNA were temporarily relaxed from p <0.05 to p <0.1 to obtain more results." 
    - I do understand the meaning of the p-value and that the thresholds can be manipulated. However, the whole dataset should be processed with the same settings, or the threshold manipulation should be justified and explained in more detail, including references that would support that. You could easily lower the threshold to 0.5 and get "more results". 

Response: I am sorry that I may have misunderstood your initial comment. The details situation here is: for the analysis of roots, all data were obtained by p < 0.05. However, no results were obtained in leaves by using this threshold. Therefore, we relax the threshold value to p < 0.1 to construct ceRNA network in leaves. We have corrected this part of the paper (line 247) to " P threshold can be temporarily relaxed in some specific cases, such as the one-sided significance level was set at 0.2 in the study of Erlotinib alone or with Bevacizumab as first-line therapy in patients with lung cancer [43]. For leaves, since no ceRNAs had been obtained in leaves by p < 0.05, the criteria of DEmRNA and DEncRNA were temporarily relaxed from p< 0.05 to p < 0.1 to explore potential ceRNAs, and qPCR was then used to verify the co-expression of all ceRNAs to exclude possible false positive results.". An appropriate reference was also added here. We apologize for the " manipulating the threshold to get more results" question due to our inappropriate description. The purpose of the threshold relaxation is not to achieve "more results" but to explore potential ceRNA networks. In addition, in Table S2, we provided the FPKM and p value of all lncRNAs, the later researchers can set the p value by themselves for screening to obtain the information they need.

  • Addition experiments

"However, due to the time needed to purchase vitamin E standard, we need an extra week to complete this experiment."
The authors require one more week to provide supporting experiments that would provide real evidence for the transcriptomic-based predictions. I believe that this extension should be allowed.

Response: I'm very grateful to you for this question. I got an unexpected benefit from your question.

The content of VE in leaves has been added to Table 1.

We found that vitamin E content was down-regulated under salt stress. In order to eliminate the erroneous results caused by problems with samples, we additionally tested VE content in samples frozen in different years (2018, 2019,2020), and the results showed that the fold-change (ST/CK) of VE content was basically consistent between samples in different years (2018,2019 and 2020). Therefore, our results on the determination of vitamin E content are reliable. This result leads to a new question: “Why the expression of tocopherol cyclase gene up-regulated while the content of tocopherol was decreased in sugar beet under salt stress?” We don't think this unexpected result is bad news. As you know that tocopherols function by reacting themselves with free radicals, which are then oxidized into α- tocopherol quinone or α- tocopherol hydroquinone. So tocopherol can actually be likened to a consumable that performs antioxidant functions in cells, which means cells can avoid damage to the membrane lipids by consuming tocopherol. Another interesting phenomenon was that the content of malondialdehyde (MDA) in sugar beet did not change significantly under salt stress. While MAD is an indicator to evaluate the degree of membrane lipids peroxidation. So we hypothesized that sugar beets protect their membrane lipids form peroxidation by producing copious amounts of tocopherol. This may be a particular mechanism by which sugar beets are more salt-tolerant than many other crops. Another piece of evidence that can be used as a reference is the tocopherol content in Oryza sativa (2.41 mg/kg), Zea mays (0.0096 mg/g), Glycine max (0.0197 mg/g) and Canola-Brassica napus (0.09 mg/g) published on Crop Composition Database (https://www.cropcomposition.org/), these data showed that tocopherol content in beet leaves (0.223 mg/g in ck) were significantly higher than in these crops. Some reports also indicated that exogenous use of vitamin E or over-expression of tocopherol cyclase gene can improve plant salt tolerance, and these studies support our conjecture.

Since sugar beets can synthesize tocopherol by themselves, the amount of tocopherol in cells should be dynamic data (the content of tocopherol =synthetic - consumption). Thus, the down-regulated of VE content under salt stress in this study may be caused by the synthesis rate of tocopherol was lower than the consumption rate of tocopherol under a high concentration (300 mM) salt stress. Additional experiments to reduce the concentration of the treatment may verify this hypothesis.

We believe that this is an issue worthy of further study in the future. Here, we added the following in the discussion section (line 364) “Under salt stress, the content of tocopherol was significantly decreased in beet leaves, while the content of MDA did not change significantly (Table 1). We hypothesized that sugar beets protect their membrane lipids from peroxidation by consuming tocopherol.”.

If I have any unclear description, please do not hesitate to contact us. Thank you again for your question

  • Spatial distribution - the authors have probably misunderstood my comment - by pooling samples from different locations, the spatial distribution within the leaf tissue is lost. The same results would have been obtained by grinding the whole leaf tissue in liquid nitrogen and aliquoting. The additional stress (which occurs within µs, see e.g. https://science.sciencemag.org/content/361/6407/1068?iss=6407) that could be relevant for the presented results would be eliminated. However, this is only a minor issue and does not need to be addressed.

Response: Thank you for your suggestion, we will update the sampling scheme in future studies.

Please, extend the figure legend of Fig. S3 to make it more self-explanatory - the fact that the whole root segment was collected is not intuitive 

Response: Legend has been added for figure s3 as required. “Fig. S3. Sampling location. Left: the third-pair of euphylla from six plants were taken and mixed for leaf sequencing. Right: a length of 6 cm from the root tip was collected from six plant samples and mixed for root sequencing.”